# Mechanism of Sand Cementation with an Efficient Method of Microbial-Induced Calcite Precipitation

**DOI:** 10.3390/ma14195631

**Published:** 2021-09-28

**Authors:** Lu Wang, Shuhua Liu

**Affiliations:** State Key Laboratory of Water Resources and Hydropower Engineering Science, Wuhan University, Wuhan 430072, China; luwang@whu.edu.cn

**Keywords:** microbial-induced calcite precipitation, compressive strength, calcium carbonate content, cemented sand, microstructure

## Abstract

This paper presents an efficient method of microbial-induced calcite precipitation (MICP) for cementation of sand particles. First, the influence of initial pH value of the culture medium on the growth of bacteria was discussed. Then, the compressive strength and calcium carbonate content of cemented sand columns with different sand particle sizes were measured to indicate the cementation effectiveness. The microstructure of cemented sand columns as well as the mineral composition and distribution of calcium carbonate were characterised by means of scanning electron microscopy-energy dispersive spectrometer (SEM-EDS) and X-ray diffraction (XRD). The results showed that the urease-producing bacteria *S. pasteurii* can be cultured at the initial pH values of 7–10, while a higher pH (e.g., 11) would hinder its growth and decrease its urease activity. The injection method of MICP with high standing time can cement sand columns effectively. Small average sand particle size of sand columns and high injection cycles can facilitate the gain of compressive strength, while calcium carbonate content of sand column higher than 8% can promote the increase of compressive strength. XRD results indicate that the fine grains generated on the surface of sand particles are calcite. The distribution of calcite on sand particles’ surface is broad and uniform. First, calcite was precipitated on the surface of sand particles, and then a precipitation layer was formed, which would connect sand particles through its high enough thickness and contribute to the development of compressive strength of the whole sand column.

## 1. Introduction

The commonly used materials for bonding loose particles, such as sand, soil or tailing powders, include Portland cement, lime, gypsum, epoxy and so on. However, these materials can cause some adverse influences. For example, Portland cement is widely used for building materials, while its production is a high energy-consuming and responsible for about 8% global CO_2_ emissions [1]. Conventional cementitious materials, such as cement, lime and so on, can increase the pH value of the treated soil to alkalinity and result in adverse effects on groundwater and vegetation [2,3]. To tackle these drawbacks, it is vital to explore and develop a new alternative cementation improvement method. In recent years, microbially induced calcium carbonate precipitation (MICP), as an eco-friendly material, has been extensively investigated in both natural environments and laboratory environments [4].

MICP technology has been applied in a variety of fields. It has been demonstrated that MICP can be used to reinforce or modify the properties of sand or soil [5,6,7,8] and to improve the cohesion and strength properties of porous materials, as well as decrease their permeability [6,9,10,11,12]. MICP has also been used to reinforce or repair construction materials, such as cement-based materials and concretes [5,13,14]. In 2004, Whiffin [15] first proposed the application of MICP on the cementation of loose sand to improve its strength and stiffness. Afterwards, the feasibility of using MICP to enhance the properties of sand or soil materials was increasingly explored. However, for lots of sand cementation, after being treated using MICP, the distribution of generated calcium carbonate is non-uniform, which would decrease the strength and reduce the cementation effect. Suitable grouting speed, concentration of cementation solution and grouting process are beneficial to improve the uniformity of cemented sand column.

This paper proposed an improving grouting process of a bacterial solution and cementation solution by providing enough time for the bacterial solution in sand columns and ensuring that calcium carbonate precipitation occurred in a static water flow. This method can improve the cementation effectiveness and produce more calcium carbonate precipitation. To validate its efficiency, a series of tests were conducted. First, the influence of the initial pH value of the culture medium on the growth process of the bacteria was investigated. Then, the compressive strength and calcium carbonate content of the cemented sand columns was measured to estimate their relationship. Finally, the microstructural characteristics were analysed using SEM-EDS and XRD, based on which the cementation mechanism of sand columns was explored and discussed in depth.

## 2. Materials and Methods

### 2.1. Materials

#### 2.1.1. Bacteria and Culture Medium

The bacteria used in this paper was the *Sporosarcina pasteurii* (CGMCC No. 1.3687) strain, a type of urea-hydrolysing and non-pathogenic bacterium. The microbial lyophilised powder was provided by the China General Microbiological Culture Collection Centre. This bacterium was isolated from the soil, which was recommended to be cultured at the temperature of 30 °C in culture medium 0907. Table 1 shows the ingredients of culture medium 0907. The pH of the culture medium was adjusted to 7.0 with 1 mol/L NaOH. The cementation solution was prepared by mixing 1 mol/L calcium chloride (CaCl_2_) solution and 1 mol/L urea (CO(NH_2_)_2_) solution.

The pH value was considered to study the influence on the growth of bacteria. The pH values of culture media were adjusted to 7, 8, 9, 10 and 11 with 1 mol/L NaOH. The bacteria were cultured under different pH value conditions and their growth was reflected by their absorbance, pH value and urease activity.

After the activation of lyophilised powder of the bacteria, inoculating it in the prepared culture medium with the inoculation dosage of 0.5% by volume, and then the bacteria were aerobically inoculated at 30 °C in a shaker with a rotation rate of 150 rpm until the liquid became turbid, always after about 30 or 40 h.

#### 2.1.2. Silica Sand

The silica sand was used in this study, which was separated with different particle size ranges: (1) 0.16–0.315 mm, (2) 0.315–0.63 mm, (3) 0.63–1.25 mm and (4) 1.25–4.75 mm. Specimen sand column A was composed of 50% sand (1) and 50% sand (3), while specimen sand column B consisted of 50% sand (2) and 50% sand (4). The average particle size of specimen sand column A is smaller than that of specimen sand column B.

### 2.2. Specimen Preparation

#### 2.2.1. Preparation of the Sand Column

A syringe (50 mL) with the inner direction of 30 mm was used as the mould of the sand column specimen. The sand column specimen in the syringe was prepared with the following procedure: (1) a layer of approximately 1 cm of gauze was placed at the bottom of the syringe; (2) 30 g sand was poured into the syringe and shake for 1 min; (3) another layer of approximately 1 cm of gauze was placed at the top of the sand column. All sand columns were positioned vertically. The peristaltic pump was connected to the injection point at the top of the column. The distilled water was flushed through the sand column three times to drive away the extra air among the sand particles before test.

#### 2.2.2. Injection Programme

One cycle injection process of bacterial solution (with absorbance of 1.1 and urease activity of 30 mM urea/min) and cementation solution was described as follows: (1) 30 mL bacterial solution was pumped into the sand column sample from the top by a peristaltic pump with a flow rate of 10 mL/min. The leaked liquid was collected and re-used to be circulated into the sand column for 1 h. Then, we let the sand column sample stand for 6 h to make the full absorption for bacteria around sand particles. (2) After completing the exudation of bacterial solution, 30 mL cementation solution was pumped into the same sand column specimen from the top by a peristaltic pump with the same flow rate. When white substance was observed in the leaked liquid, the bottom flow outlet of the sand column was sealed, standing for 10 h for full reaction of the system.

### 2.3. Test Methods

#### 2.3.1. Three Indicators of Bacterial Growth

The absorbance, pH value and urease activity were selected as three indicators to measure the growth of bacteria. The absorbance was determined by measuring the optical density at a wavelength of 600 nm using a spectrophotometer (721N, INESA (Group) Co., Ltd., Shanghai, China), which was usually applied to compare different growth characteristics of the bacteria [16], recorded as OD600. The spectrophotometer was calibrated using un-inoculated growth media as blank before the optical density of bacterial cultures grown was measured [17]. Then, 3 mL bacterial solution was placed into a clean cuvette for test [17]. After the test of OD600, the bacterial solution in cuvette was placed into a 7 mL centrifuge tube to measure its pH using a pH meter (Leici PHS-3C, INESA (Group) Co., Ltd., Shanghai, China). The pH electrode was calibrated with pH 6.86 and 9.18 buffer solutions before each test. Urease activity (UA) of the bacterial solution was determined by the relative conductivity change of a mixture with 1 mL of bacterial solution and 9 mL of 1.5 M urea over 5 min at 25 °C [15]. The urease activity can be calculated as:(1)$${\mathrm{Urease}\;\mathrm{activity}\;(\mathrm{mM}\;\mathrm{urea/min})=(C}_{2}\times {10-C}_{1})/5\times 11.11\;\;(R^{2}=0.9988)$$
where C_1_ is the initial conductivity of the test bacterial solution; C_2_ is the conductivity of the bacterial solution after mixing with urea; 10 is the dilution factor of the test bacterial solution; 11.11 is the conversion coefficient from the conductivity per minute of urease activity.

#### 2.3.2. Compressive Strength

After three and five cycles of injection processes for each sand column specimen, the sand column specimens with syringe moulds were kept in a drying oven at 30 °C for 72 h. Then, the sand columns were demoulded, and both ends of the sand columns were polished for compressive strength test. The compressive strength was tested using a microcomputer controlled flexural and compression testing machine with the type of YAW-300C, and the loading speed was 0.6 kN/s. After the determination of compressive strength, the debris of the broken sand columns were collected for calcium carbonate content and microscopic analysis.

#### 2.3.3. Calcium Carbonate Content

Following the compressive strength test, the calcium carbonate content was measured according to the gravimetric acid washing method described in [18]. Approximately 15 g of the oven-dried consolidated sand (recorded as M_1_) was soaked in 1 M of hydrochloric acid (HCl), which was shaken until no gas spilled out. Then, the dissolved calcium carbonate and acid solution were passed through a cellulose filter paper and rinsed three times to wash out all ions from the sand. The residue sand on the filter paper was then oven-dried (recorded as M_2_). The difference between both M_1_ and M_2_ represents the mass of calcium carbonate, which can be expressed as:(2)$${M(\mathrm{CaCO}}_{3}{)(\%)=(M}_{1}{-M}_{2}{)/M}_{1}\times 100\%$$

#### 2.3.4. Microscopic Tests

The collected debris of sand columns were stored in a vacuum drying oven at 60 °C until the microscopic tests, XRD and SEM-EDS, were done. Small debris of sand columns were ground to pass through a 80 µm sieve to test the crystal composition. XRD analysis was employed to analyse the precipitate material and crystal type using X-ray Diffractometry (X’ Pert Pro, PANalytical B.V., Almelo, The Netherlands), with copper target. The XRD pattern was obtained by a continuous scan from 10° to 80°, with a step size of 0.02°.

SEM-EDS analysis was performed to analyse the surface topography, morphology and mineralogical compositions of the precipitate material using Field Emission Scanning Electron Microscope (Zeiss SIGMA, Carl Zeiss AG, Oberkochen, Germany), together with energy dispersive spectrometer (Oxford Ultim Max 40, Oxford Instruments, Oxford, UK), with accelerating voltage of 5 kV, magnification of 5× up to 100,000× and resolution of 1.0–2.0 nm. Note that before the SEM and EDS tests (except SEM-EDS mapping) all samples would be coated with a thin layer of gold by an Auto Fine Coater with an electric current of 30 mA and a time of 90 s, to improve the electrical conductivity of the samples’ surface.

## 3. Results and Discussion

### 3.1. The Growth of Bacteria with Different Initial pH

The culture of *S. pasteurii* needs certain conditions, as the environmental factors are very important for its culture. Herein, the pH value of culture medium was considered to investigate the growth process of *S. pasteurii* culture. The effect of growth time on the OD600, pH and UA of the bacterial solution under different initial pH values is shown in Figure 1.

As seen in Figure 1a, the change of OD600 with growth time represents the grow curve of *S. pasteurii*. The growth curves of all samples can be divided into three stages from Figure 1a: a lag phase, an exponential phase, and a stationary phase. During the lag phase, *S. pasteurii* tends to adjust to the high nutrient environment and prepares for fast growth [19]. During the exponential phase, the growth of cells becomes faster and consumes nutrients at a fast rate. During the stationary phase, the growth rate of *S. pasteurii* gets hindered due to the depletion of nutrients. When the initial pH is increased from 7 to 9, the change of OD600 of each sample follows a similar trend: first increasing slowly, followed by a quick increase and, finally, a stable stage. At an initial pH of 10, OD600 gets a slight decrease compared to the former three samples at every growth time. However, when the initial pH reaches 11, OD600 becomes close to 0 after 7 h of growth and then increases while the increasing rate is much lower than others. The final value of OD600 is about 0.7, which is the lowest value. This can be ascribed to the high alkalinity in the culture medium, which can hinder the growth of *S. pasteurii*.

As seen in Figure 1b, the pH value of each sample was recorded with the increase of growth time. No matter what the initial pH value of the sample is, the final pH value of all samples is about 9.5. When the initial pH is lower than 9.5, with the growth of *S. pasteurii*, some alkaline elements moved into the system, leading to an increase of its pH. When the initial pH is higher than 9.5, the high alkalinity in culture medium could hinder the growth of *S. pasteurii*, and some OH^−^ was consumed during this period, resulting in a reduction of the pH value.

Figure 1c shows the change of UA with a growth time of *S. pasteurii* under different initial pH values. When the initial pH rises from 7 to 10, the UA values of all samples follow an almost same changing trend, which are increased with the growth time and reach a stable value at last. When the initial pH is 11, the UA has almost no change at the first 7 h growth time, followed by a gradual increase until a stable value of 22 mM urea/min is reached, which is much lower than that for others, i.e., approximately 33 mM urea/min.

The initial pH of culture medium has a significant influence on the growth process of *S. pasteurii*. The higher alkalinity of culture medium could hinder the growth of *S. pasteurii*, which would reduce the OD600 and UA. When the initial pH value is within the range of 7 to 9, there is no influence of OD600 and UA on the growth process of *S. pasteurii*, while with the increase of the initial pH value from 10 to 11, the decreasing effect of OD600 and UA becomes obvious. Especially for the initial pH value of 11, the OD600 and UA value decrease significantly in comparison with other samples. Thus, it is indicated that the high alkaline condition (e.g., a pH value of 11) is not suitable for the growth of *S. pasteurii*.

### 3.2. Compressive Strength of Cemented Sand Column

Figure 2 shows the compressive strength of the cemented sand columns after three and five injection cycles of bacterial solution and cementation solution. As the injection cycles increase from three to five, the compressive strength of each sand column increases. The compressive strength of specimen sand column A is increases by 50.72% from 1.38 to 2.08 MPa, while the compressive strength of specimen sand column B is increased by 61.11% from 0.72 to 1.16 MPa. The increase of compressive strength of each sand column by more than 50% with increasing injection cycles from three to five indicates that after five injection cycles can promote the gain of compressive strength. Compared to specimen sand column B, the specimen sand column A has a higher compressive strength, regardless of the injection cycles. As introduced in Section 2.1, the sand particle size of sand column A is smaller than that of sand column B. The space between sand particles of sand column A is small and, thus, the space for MICP between sand particles is small. Hence, at the same injection cycles, the required less calcium carbonate is required for filling in the space between sand particles, resulting in a faster filling of MICP in the space between sand particles and then a gain of compressive strength. When the injection cycle is higher, more calcium carbonate would be generated to fill in the space between sand particles and, therefore, the compressive strength would be higher. The injection cycles and average sand grain size play an important role in the compressive strength of bio-cemented sand column. It was reported by Yu [20] that the compressive strength of bio-sandstone with sand particle range of 0.212–0.525 mm after two, four and six injections of bacterial solution and substrate solution is 0.37, 0.63 and 1.33 MPa, respectively. A highest compressive strength of 0.58 MPa of sand column can be obtained, which is much higher than that of untreated sand with the same density, i.e., below 0.20 MPa [21]. In our study, the compressive strength of sand column A with sand particle range of 0.16–0.315 and 0.63–1.25 mm is 2.09 MPa with five injection cycles, which is much higher than that reported in [20] of 1.33 MPa with six injections. Moreover, the strength of sand column B with a sand particle size of 0.315–0.63 and 1.25–4.75 mm is 0.72 MPa with three injection cycles, which is also higher than that reported in [20] of 0.63 MPa with four injections. Although there are differences in these two experiments, the strength of the sand column with a high average particle size and lower injection cycles in our study is still higher than that reported in [20], which is likely to be attributed to the different injection methods used, suggesting that our injection method is more efficient. Multiple injections of bacterial solution and substrate solution can facilitate the gain of compressive strength of bio-sandstone. A smaller average particle size of sand column can lead to a more efficient gain of compressive strength.

### 3.3. Calcium Carbonate Content of Cemented Sand Column

The calcium carbonate content of each sand column with three and five injection cycles is shown in Figure 3. The content of calcium carbonate in the sand column increases with the increase of injection cycles from three to five cycles. The calcium carbonate content in sand column A is increased by 34.31% from 7.5% to 10.07%, while the calcium carbonate content in sand column B is increased by 61.93%, from 5.12% to 8.26%. This indicates that increasing the injection cycles is beneficial to the increase of the calcium carbonate deposition content in the sand column. The content of calcium carbonate in sand column A is higher than that in sand column B at each injection cycle. The average particle size of sand in sand column A is smaller than that in sand column B and less empty space exist between the sand particles in sand column A, as a result of which less calcium carbonate is required to fill the space between sand particles. Therefore, at the same injection cycles, sand column A can achieve a better filling effect, and the calcium carbonate generated by its deposition can fill the space between sand grains faster.

The relationship between the calcium carbonate content and compressive strength of the sand columns is shown in Figure 4. With the increase of the calcium carbonate content, the compressive strength of the sand column is increased. When the calcium carbonate content is less than 8%, the increment of compressive strength is small with increasing calcium carbonate content, which is consistent with the finding presented in [22], which showed that the low calcite content did not significantly improve the compressive strength of the sand column samples. When the calcium carbonate content is higher than 8%, a significant increment of compressive strength with the increase of calcium carbonate content can be found. This agrees with the finding for bio-sandstone that the sample with the maximum amount of CaCO_3_ had the highest compressive strength [22]. Five injection cycles were found to be efficient to gain compressive strength of the sand column. Whiffin [9,15] demonstrated that there is a good correlation between the UCS of the sand column and the calcite content, that is, when the calcite content is less than 3.5% or 60 kg/m^3^, the compressive strength of the sand column has no obvious improvement but is enhanced significantly when the calcite deposition is more. According to the change trend between the calcite content and compressive strength, the correlation between them may conform to an exponential function, shown in Equation (3). The exponential relationship between the compressive strength and calcite content is in line with that reported in the literature [23,24,25,26,27]:(3)$$S=0.1628\times e^{0.2450x},\;R^{2}=0.9423$$
where S is the compressive strength and *x* is the percentage of calcite content.

### 3.4. Sand Cementation Mechanism

Figure 5 displays the microstructure of cemented sand columns with different injection cycles. As seen in Figure 5a of the microstructure of sand column A with three injection cycles, a lot of fine grains exist among the intersection of sand particles and the size of fine grains is about 2–5 µm with irregular shape. Some pores among sand particles can be observed. Figure 5b shows the microstructure of sand column A with five injection cycles, where more irregular shape grains can be found around the sand particles. Obviously, a layer of generated substance coats the surface of sand particles, which are cemented with each other, and thus, the whole sand body gains compressive strength. Moreover, the pores between sand particles are smaller than that of sand column A with three injection cycles, suggesting that more injection cycles would generate more calcite crystals and fill in more pore space. With the increase of injection cycles, the number of pores of the cemented sand column reduces so that the inner structure of the cemented sand column is being compacted step by step [28]. Figure 5c illustrates the microstructure of sand column B with three injection cycles, indicating that although some fine grains are generated, there are not enough grains to fill in the space among sand particles. In addition, the average sand particle size of sand column B is larger than that of sand column A, the pore space among sand particles is larger. As a result, more fine grains are required to fill the pores; on the other hand, the generated fine grains are more easily washed away during the process of injection. Figure 5d shows the microstructure of sand column B with five injection cycles. A layer of substance can be found on the surface of sand particles, and the pore space among sand particles is filled with fine grains, while some interspaces among the fine grains can be found, which is the weak point.

To analyse the chemical composition of the fine grains generated on the surface of sand particles, EDS analysis of sand column A with 5 injection cycles was conducted and the results are shown in Figure 6. Clearly, the main chemical compositions of the irregular shape substance include calcium (Ca), carbon (C) and oxygen (O), implying that the generated fine grains belong to calcium carbonate.

According to the literature, the crystal types of calcium carbonate are multitudinous, including calcite, vaterite, aragonite and amorphous calcium carbonate, and different crystal types of calcium carbonate possess different properties [29]. For example, calcite is the most stable among them. Effective crystals are those which precipitate in the pores of the soil and bond soil particles to each other like a bridge, to tolerate the imposed load between soil particles and enhance the specimen’s strength [30]. Thus, to clarify the crystal type of the generated calcium carbonate is very important, as some sand particles, including the generated calcium carbonate on its surface, were ground to powder for the XRD analysis. The XRD patterns of the samples with different sand columns and injection cycles are presented in Figure 7. From Figure 7, for every specimen the intensity of silica characteristic peaks are significantly high, which is because the main composition of sand particles is silica; thus, the results are predictable. Otherwise, the diffraction peaks of calcite are different for every specimen. It can be indicated that calcite is the fine grains generated on the surface of sand particles in Figure 5. The intensity of the diffraction peak of calcite increases with the increase of injection cycles from three to five cycles and the decreases with the increase of sand particle size from small to large. The intensity of diffraction peak can reflect the content of the detected substance. Therefore, from Figure 7, calcite content of sand column A with five injection cycles is the highest, followed by sand column B with five injection cycles, sand column A with three injection cycles and sand column B with three injection cycles, for which the changes in compressive strength and calcite content are described in Section 3.2 and Section 3.3.

To analyse the distribution of the generated calcite around sand particles, a debris of sand column A-5 without a gold coat was tested by means of SEM-EDS with mapping, the result of which is presented in Figure 8. The distribution of elements C, O and Ca is almost identical, suggesting that the substance distributed in these areas is calcium carbonate. At the same time, the distribution of elements of O and Si is almost identical as well. SiO_2_ exists at the same distribution areas. The distribution of Ca can represent the distribution of calcium carbonate. As seen in the EDS mapping, the distribution of calcium carbonate is broad and uniform.

According to the results above, after the injection treatment of sand columns, lots of calcite crystals are generated around the surface of sand particles, coating the sand particles and taking a role of bridge function between adjacent sand particles. For untreated sand columns, there exists a lot of interspace between the sand particles. A bacterial solution is firstly introduced into the sand column from the top, which then stands for 6 h to make sure that more bacteria could be absorbed on the surface of sand particles. Afterwards, the cementation solution is introduced into the sand column. With the presence of the urea and calcium source, the calcite crystals are produced. With an increase of injection cycles, the absorbed bacteria on the surface of sand particles would increase, leading to the formation of more calcite. The metabolism of microorganisms provides alkaline conditions for the system. Biofilms and extracellular polymers formed by bacterial secretion can effectively bind ions in the surrounding environment and becomes heterogeneous nuclear sites for mineral deposition. Crystal nucleation begins to produce active sites by binding various cations, while functional groups, such as carboxyl, hydroxyl and phosphate, remove the proton from bacterial cell walls under alkaline conditions. Therefore, strong electrostatic attraction with cations occurs, and calcium ions gather on the surface of the cell wall, eventually facilitating calcium ions to combine with carbonate ions to form calcite. When calcite reaches the supersaturation state, it will precipitate out on the surface of sand particles. The injection treatment, especially for the standing period, gives enough time for the formation of calcite crystal. Calcite crystal first precipitates on the surface of sand particles, and then a precipitation layer around sand particles is formed. After that, with the increase of the precipitation layer thickness, the sand particles that are closer together would be connected by the precipitation layer, converting the loose sand column into a whole sand body with a certain degree of compressive strength [21,31]. It was illustrated that calcite of equal thickness which precipitated on the surface of sand particles offers a weak bonding effect, which cannot improve the soil properties, while the calcite formed at the intersection of sand particles is beneficial to the property enhancement of soils [10]. As bacteria is easy to adsorb on small surfaces, e.g., the intersections among sand particles, higher concentration of bacteria leads to more calcite deposition in this area. On the other hand, the calcite deposited on the surface of the pore solution and soil particles would be adsorbed near the intersections of particles as the solution flows through the pore throat [32]. Therefore, the calcite precipitation on the intersections of sand particles is more than that at other places, while it cannot take out the contribution of calcite precipitated on the surface of sand particles to compressive strength.

When the injection cycle is increased from three to five, more bacteria could be absorbed on the surface of sand particles, resulting in more calcite precipitation to fill in the pores among sand particles. As shown in Figure 5, more pores exist among sand particles of sand column with three injection cycles compared to that with five injection cycles. At the same time, when the pore is greater, the generated calcite is easier to be washed out during the injection process. The average sand particle size of sand column B is larger than that of sand column A, so the voids among sand particles of sand column B are much greater than that of sand column A and the produced amount of cementation agent per injection during a MICP process is limited [33]. The greater the pores, the easier the generated calcite to be washed away. As seen in Figure 5a,c, there are more and larger pores in sand column B compared to sand column A. Because the cementation of the sand column depends on the connection of the precipitation layer between sand particles, the more there is calcite precipitation, the more effective the connection of the sand column. It can be observed from the SEM graphs that the connection of the precipitation layer between sand particles of sand column A is better. The compressive strength and calcite content also support this finding. As the average sand particle size is small, the generated pores among sand particles are also small. Fine sand particles may provide more effective nucleation sites for calcite precipitation [21]. This also results in a larger surface of sand particles to absorb more bacteria and, thus, more calcite precipitation would be generated and connected, which contributes to the compressive strength of sand column.

## 4. Conclusions

According to the results above, an effective injection process has been proposed to cement sand columns, after which a series of tests were conducted to validate its efficiency and the cementation mechanism of sand columns was explored and discussed. The following conclusions can be draw:

(1) Urease-producing bacteria *S. pasteurii* can be cultured at the initial pH values of 7–10, while a higher pH (e.g., 11) would hinder its growth and decrease its urease activity.

(2) The smaller average particle size of sand particles of sand column and the higher injection cycles would be more effective to promote the gain of compressive strength. When the calcium carbonate content of a sand column is higher than 8%, it more efficiently increases compressive strength.

(3) XRD results indicate that the fine grains generated on the surface of sand particles are calcite, and the intensity of diffraction peak can increase with the decrease of the sand particle size and the increase of the injection cycles.

(4) The distribution of calcite on sand particles’ surface is broad and uniform. First, calcite is precipitated on the surface of sand particles, and then, a precipitation layer is formed, which will connect sand particles because it has high enough thickness and will contribute to the compressive strength of the whole sand column.

## Figures and Tables

**Figure 1 materials-14-05631-f001:**
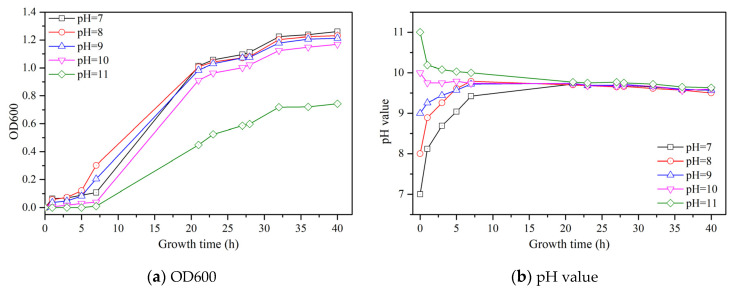
Effect of growth time on the OD600, pH and UA of bacterial solution under different initial pH values.

**Figure 2 materials-14-05631-f002:**
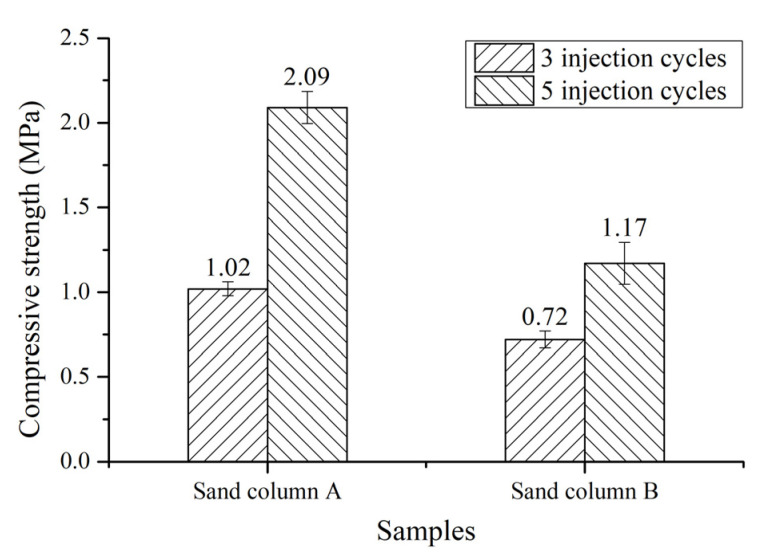
Compressive strength of the cemented sand columns.

**Figure 3 materials-14-05631-f003:**
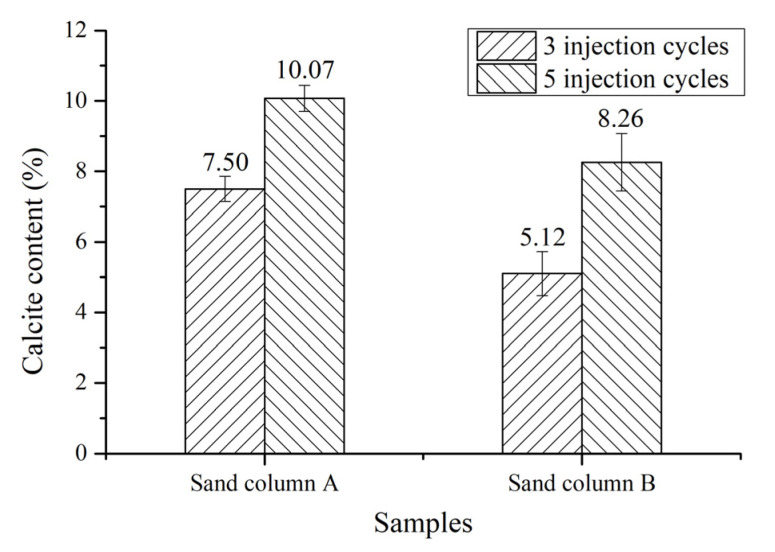
Calcium carbonate content of each sand column.

**Figure 4 materials-14-05631-f004:**
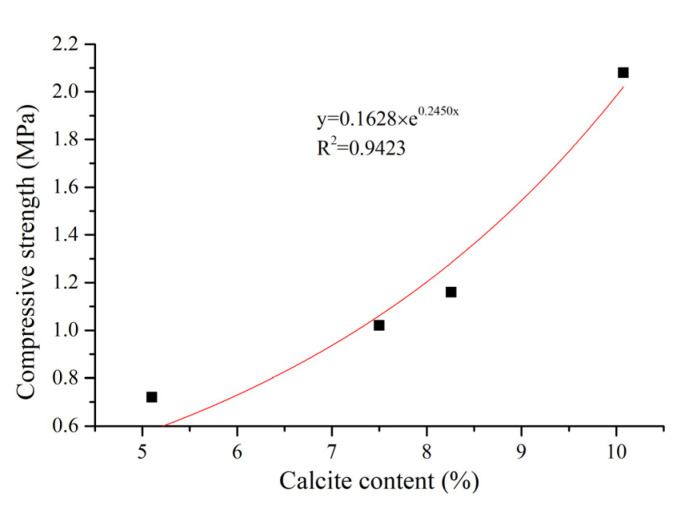
Relationship between calcium carbonate content and the compressive strength of sand columns.

**Figure 5 materials-14-05631-f005:**
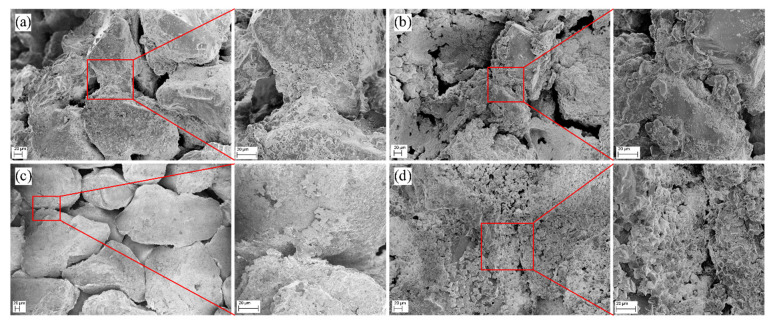
Microstructure of cemented sand column A with (**a**) three injection cycles and (**b**) five injection cycles; microstructure of cemented sand column B with (**c**) three injection cycles and (**d**) five injection cycles.

**Figure 6 materials-14-05631-f006:**
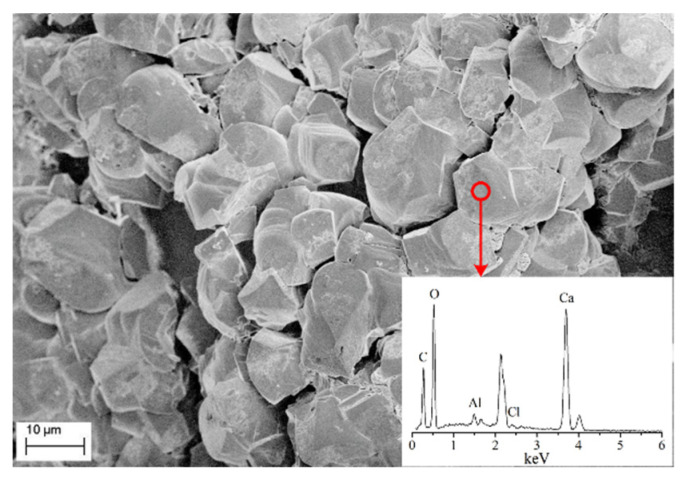
SEM-EDS results of the irregular shape grains.

**Figure 7 materials-14-05631-f007:**
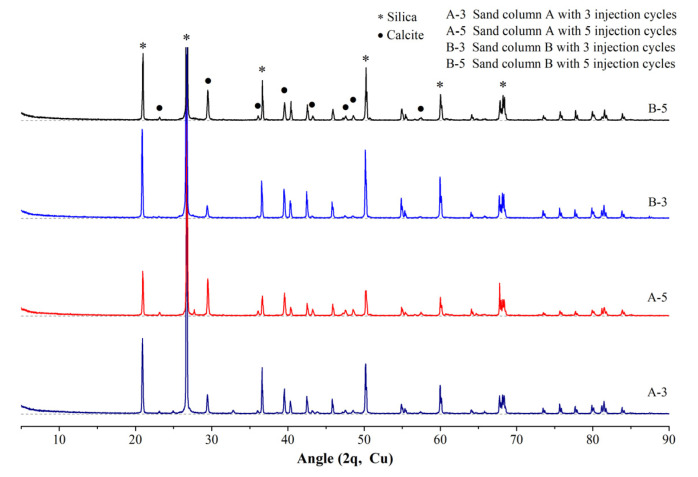
XRD results of cemented sand columns.

**Figure 8 materials-14-05631-f008:**
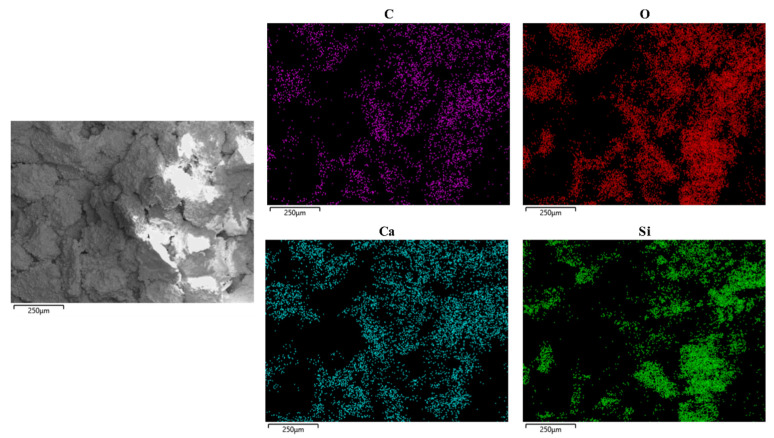
SEM-EDS mapping of the sand column.

**Table 1 materials-14-05631-t001:** Ingredients of culture medium 0907.

Bacteria Type	Ingredients of the Culture Medium 0907 (L^−1^)				
	Peptone	Beef Extract	Urea	MnSO_4_·H_2_O	Agar
Sporosarcina pasteurii	5 g	3 g	20 g	0.01 g	15 g

## Data Availability

The data presented in this study are available on request from the corresponding author.

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
