# Peer review of "Mechanism of Sand Cementation with an Efficient Method of Microbial-Induced Calcite Precipitation"

_materials, 2021, doi:10.3390/ma14195631_

Round 1

Reviewer 1 Report

The present manuscript reports an interesting study, with an organized and fluid reading. The proposed approach revealed quite valuable results towards effective strategy of microbial induced calcite precipitation. Of special interest is the fact, that the Authors proposed and described a mechanism governing san cementation. Nevertheless, there are some minor issues to be clarified, as follows:

  1. Section 2.1.1. – How long was the culture? For how many days, hours did you culture bacteria? It can be concluded from the text of the manuscript, that the culture time was set to 40 hours. If so, please consider the addition of this information in the aforementioned section of the manuscript. It would be more informative for the Readers.
  2. Section 2.3.4. – Were the samples coated with a conducting agent prior to SEM imaging?
  3. Section 3.1. – The pH influence on pasteurii is known and was already described in the literature, thus in my opinion, this section brings nothing new. Please comment on that.
  4. Section 3.2. - The Authors stated that: “The compressive strength obtained in this study is slightly higher than that reported in [33], which can be attributed to the different injection methods used, suggesting that our injection method is more efficient.” (Page 7, Lines 28-20).

However it should also be noted by the Authors that the experiment conducted by Yu and coworkers differs from the experiment described in the revised manuscript by other factors, like average particle size of sand, which can also influence mechanical test results.

  1. Section 3.4. (Figure 5c) – Why the scalebar is set to 200 µm, while all other images are described with 20 µm scalebar? I presume this is a typo, please correct it. If not, please provide an image with 20 µm scalebar to fit the other images.
  2. Section 3.4. (Figures 6 and 8) – Please specify which sample was measured? Were the same results acquired for other samples? In addition please comment, how did you choose the area for SEM investigation. Was the cemented microstructure observed in a specially selected area, or was it a randomly selected area?

Reviewer 2 Report

The paper is original and interesting.

I have only few questions and suggestions:

  • in abstract there is lack of more information about results - presentation some most important values; in current form abstract contain only general informations without any important values;
  • what was the criteria the choose of absorbance, pH value, and urease activity as a indicators to measure the growth of bacteria ? 
  • please check references style - in some place there are "pp" and in some no;

Reviewer 3 Report

The article reproduces the well-known effect of bacteria-induced biocementation. It is usually used to strengthen soils and reduce the flowability of sands. Only some of the details of the experiment carried out can be considered original in the work: the use of a specific type of microorganisms, the choice of fractions of the used sand and the method of their impregnation. The work is of an applied nature and to a greater extent corresponds to the topics of journals: Journal of Geotechnical and Geoenvironmental Engineering, Construction and Building Materials and others related to geochemistry and building materials.

Round 2

Reviewer 3 Report

It makes sense for the authors to justify the use of a specific type of microorganism in such experiments